# Multidisciplinary Care of Patients with Facial Palsy: Treatment of 1220 Patients in a German Facial Nerve Center

**DOI:** 10.3390/jcm11020427

**Published:** 2022-01-14

**Authors:** Jonathan Steinhäuser, Gerd Fabian Volk, Jovanna Thielker, Maren Geitner, Anna-Maria Kuttenreich, Carsten M. Klingner, Christian Dobel, Orlando Guntinas-Lichius

**Affiliations:** 1Department of Otorhinolaryngology, Jena University Hospital, 07747 Jena, Germany; jonathan.steinhaeuser@live.de (J.S.); fabian.volk@med.uni-jena.de (G.F.V.); jovanna.thielker@med.uni-jena.de (J.T.); maren.geitner@med.uni-jena.de (M.G.); anna-maria.kuttenreich@med.uni-jena.de (A.-M.K.); christian.dobel@med.uni-jena.de (C.D.); 2Facial Nerve Center Jena, Jena University Hospital, 07747 Jena, Germany; carsten.klingner@med.uni-jena.de; 3Center for Rare Diseases, Jena University Hospital, 07747 Jena, Germany; 4Department of Neurology, Jena University Hospital, 07747 Jena, Germany

**Keywords:** multidisciplinary care, Bell’s palsy, chronic facial palsy, facial paralysis, facial nerve surgery, botulinum toxin, physiotherapy

## Abstract

To determine treatment and outcome in a tertiary multidisciplinary facial nerve center, a retrospective observational study was performed of all patients referred between 2007 and 2018. Facial grading with the Stennert index, the Facial Clinimetric Evaluation (FaCE) scale, and the Facial Disability Index (FDI) were used for outcome evaluation; 1220 patients (58.4% female, median age: 50 years; chronic palsy: 42.8%) were included. Patients with acute and chronic facial palsy were treated in the center for a median of 3.6 months and 10.8 months, respectively. Dominant treatment in the acute phase was glucocorticoids ± acyclovir (47.2%), followed by a significant improvement of all outcome measures (*p* < 0.001). Facial EMG biofeedback training (21.3%) and botulinum toxin injections (11%) dominated the treatment in the chronic phase, all leading to highly significant improvements according to facial grading, FDI, and FaCE (*p* < 0.001). Upper eyelid weight (3.8%) and hypoglossal–facial-nerve jump suture (2.5%) were the leading surgical methods, followed by improvement of facial motor function (*p* < 0.001) and facial-specific quality of life (FDI, FaCE; *p* < 0.05). A standardized multidisciplinary team approach in a facial nerve center leads to improved facial and emotional function in patients with acute or chronic facial palsy.

## 1. Introduction

Peripheral facial palsy is the most frequent cranial nerve palsy causing significant functional and psychological morbidity. The management of patients with facial palsy can be challenging because there are over 50 etiologies [1]. The annual incidence of idiopathic Bell’s palsy as the most frequent type of acute facial palsy is reported to be 20 to 40 of 100,000 persons [2]. As about 70% of the cases of acute facial palsy are Bell’s palsy, the overall incidence of acute facial palsy is 29 to 57 of 100,000 persons per year [3]. Depending on the severity of the lesion, at least 30% of the cases will not recover completely. These cases will remain flaccid (chronic flaccid facial palsy) if not treated otherwise or develop post-paralytic synkinesis due to pathological facial nerve regeneration. There is a broad and continuously evolving spectrum of diagnostic and management approaches for facial palsy [4]. The management of patients with facial palsy often requires complex clinical decision-making [5]. Optimal diagnostics are needed to detect early the non-idiopathic cases, cases with worse prognosis, or candidates for immediate or early reconstruction surgery [1]. Although several clinical guidelines have been published, many cases still do not receive optimal treatment. Moreover, patients with a low probability of recovery are not referred or referred very late to specialized facial palsy services [2,6].

The multidisciplinary centralized approach for facial palsy in a referral center has several advantages. Treatment in a specialized center allows a systematic and up-to-date assessment of facial function, functional and psychological considerations, assessment of quality of life, elaboration of guidelines, emotional support, comprehensive long-term care, and facilitated inclusion into clinical trials [5,7]. As such, a series of 3650 patients treated in Pittsburgh between 1963 and 1996 and a series of 1989 patients treated in Boston between 2003 and 2013, with a focus on decision-making but not on the outcome, have been published [4]. A multidisciplinary collaboration, including a wide variety of subspecialties, has proven effective for the treatment of patients with Bell’s palsy [8]. A patient-centered approach, utilizing physiotherapy, targeted botulinum toxin injections, and selective surgical intervention offered by a multidisciplinary team can effectively reduce the burden of long-term disability for patients with Bell’s palsy and longstanding sequelae [8]. The same has been shown for the management of facial paralysis following skull base surgery [5]. These patients profit from a multidisciplinary intervention because an individualized combination of pharmacologic therapy, physical therapy for facial neuromuscular retraining, and surgical intervention is needed for most patients [5]. Krane et al. reported on establishing a facial nerve center based on the experience with 22 surgical cases treated between 2014 and 2019 [9]. Of particular note is that not only standard facial grading but also facial-nerve-specific patient-related outcome measures (PROMs) were used to measure the outcome in a standardized manner. In doing so, Krane et al. could show that nerve transfer and free gracilis muscle transfer not only improved smile excursion and facial symmetry but also the quality of life of these patients [9]. The most recent report comes from the Sydney Facial Nerve Clinic on 145 patients treated between 2015 and 2018 [10]. The Sydney team also used both classical facial grading and PROMs for the initial assessment of the patients and during follow-up. This allowed them to show that it is not the physical level of function that a patient has but the social and psychological impact of their palsy that drives them to presentation in a specialized facial nerve center [10]. Another more historical but large (>1000 patients) series focused on Bell’s palsy [11,12,13]. These historical series do not include standardized outcome measures and are limited to the measurement of the physical dysfunction of the patients. These studies do not include data on the quality of life of the patients or their psychosocial dysfunction.

When assessing the outcome of a facial nerve center, it seems to be advisable to measure the outcomes with standardized grading tools and PROMs. This is of interest for patients with acute and chronic facial palsy. Therefore, we aim to report the experience from a German multidisciplinary facial nerve center, treating patients with acute and chronic facial palsy. The focus lies on the diagnostic and therapeutic management of 1220 patients. The patients are not only initially assessed and later on monitored by facial nerve grading but also from the beginning by facial nerve specific (PROMs). This allows us to critically analyze the outcome in our center. Furthermore, we can compare the results to other multidisciplinary centers using a similar approach.

## 2. Material and Methods

### 2.1. Ethical Considerations

The study was conducted according to the guidelines of the Declaration of Helsinki and approved by the Ethics Committee of the Jena University Jena, Germany (protocol code 4665-01/16; approved at 22 January 2016).

### 2.2. Study Design and Inclusion Criteria

This observational cohort study was based on the database of the Facial Nerve Center, Jena University Hospital, Jena, Germany. Standardized prospective data collection on all patients admitted with facial palsy was started in 2007. The patients were admitted to the Department of Otorhinolaryngology or the Department of Neurology of the Jena University Hospital, Jena, Germany. The database includes patients’ baseline characteristics, all diagnostics, and data of all outpatient, daycare, and inpatient treatments. The multidisciplinary team approach was started in 2008. The team also includes a psychologist, speech therapists, and physiotherapists to see all new patients. The Facial Nerve Center in Jena was formally established in 2012. Here, we present all patients admitted with facial palsy (International Classification of Diseases [ID] codes G51.0, G51.1, G51.2, G51.8, and G51.9) between 2007 and 2018. The inclusion was limited to 2018 to allow an adequate follow-up. Only patients with hemifacial spasm (G51.3), facial myokymia (G51.4), and facial dystonia (G24*) were excluded. Otherwise, the study ought to reflect the complete health care service of the center for patients with any kind of facial palsy. Most patients came from Germany (99.3%). Over the years, the number of referred patients has increased continuously, mainly from other federal states in Germany (Appendix A). The diagnostic and therapy approach was different for patients referred with acute facial palsy (≤90 days after onset; acute palsy group) and patients with chronic facial palsy (>90 days after onset; chronic palsy group) and depending on the underlying disease. In the chronic phase, patients were either referred with flaccid facial paralysis/flaccid paresis of parts of the face or post-paralytic facial synkinesis. Some patients were referred with acute or chronic flaccid paralysis, became patients with post-paralytic synkinesis after nerve repair, and were eventually also treated for synkinesis (Appendix A).

### 2.3. Diagnostics and Assessment of Initial and Final Facial Nerve Function

Depending on the etiology of acute facial palsy, or if unknown, a battery of diagnostics is performed. Basic diagnostics includes otorhinolaryngologic examination, including ultrasound of head and neck and a neurological examination, laboratory tests, pure tone audiometry, vestibular function tests, the stapedius reflex test, the gustatory test, and the Schirmer test [14,15]. Baseline electroneurography, blink reflex testing, and needle electromyography (EMG) were performed as fast as possible in case of acute facial palsy [16]. If the first EMG was performed earlier than 14 days after onset, the examination was repeated at least once and later than 14 days after onset.

In the case of chronic facial palsy, EMG was also the central electrodiagnostic test to confirm complete denervation in chronic flaccid palsy or to confirm synkinetic reinnervation in the case of pathological reinnervation with post-paralytic synkinesis. Facial muscle ultrasound was introduced in 2011 and has been a routine diagnostic tool since 2013 in long-term flaccid facial palsy to analyze the viability of the facial muscles [17]. The most recently implemented diagnostic tool was diagnostic electrostimulation in 2015. This enables a direct electrophysiological evaluation of long-term denervated facial musculature [18,19].

At each presentation, the facial palsy is graded according to the Stennert index [20]. The Stennert index classifies the face at rest (0–4 points; 0 = normal to 4 = complete loss of resting tone) and during motion (0–6 points; 0 = normal to 6 = no motion) separately. The points for both scores are summarized to a total score (0 = normal; 10 = worst dysfunction). Clinically, the palsy is defined as complete if the patient presents with a complete loss of motor function in the affected hemiface or if the palsy has deteriorated to a complete palsy during the inpatient course of treatment. Otherwise, the palsy is defined as incomplete palsy. The degree of facial nerve dysfunction is not classified as “paresis” = incomplete loss of facial nerve function and “paralysis” = complete loss of facial nerve function. Instead, we use here the umbrella terms “palsy” or “facial nerve dysfunction”. The patient’s perspective was regularly assessed using the German versions of two patient-reported outcome measures (PROMs). The Facial Clinimetric Evaluation (FaCE) scale and the Facial Disability Index (FDI) were used [21,22,23]. The FDI questionnaire comprises 10 Likert-type questions, divided into two domains, and includes physical function and social/well-being function. The physical function scale is scored from −25 (worst) to 100 (best). The social/well-being function scores range from 0 (worst) to 100 (best). Both FDI scales are summed to a total score. The FaCE has six independent domains: social function, facial movement, facial comfort, oral function, eye comfort, lacrimal control, and a total score incorporating all domains. Each FaCE score ranges from 0 (worst) to 100 (best). The 36-item SF-36 Health Survey (SF-36) measures the general quality of life [24]. Outcome criteria for all patients were the absolute change of the Stennert index and the PROMs between their first and last visits to the Facial Nerve Center.

The evaluation of patients with post-paralytic synkinesis included a psychological assessment of the suitability for in-house daycare EMG biofeedback training. This assessment contained the Body Dysmorphic Disorder Munich Module (BDD-MM) [25], the Beck Depression Inventory (BDI) [26], and the Liebowitz Social Anxiety Scale (LSAS) [27]. Results of this assessment were published recently and will not be presented here [28].

### 2.4. Treatment

In the case of acute facial palsy, the underlying disease directed the treatment under consideration of the German guideline for the treatment of facial palsy [29,30]. As symptomatic treatment and in each case of idiopathic facial palsy, a tapered course of corticosteroids over 7 days is regarded as standard treatment [30]. In the case of VZV reactivation, but also for many patients with idiopathic facial palsy, the patients received additional acyclovir for 5 days. Ceftriaxone was given for 7 days for patients with Lyme disease (mainly stage 2, early disseminated infection; seldom stage 3, late persistent Lyme disease).

The center offers a spectrum of surgical and non-surgical treatments, mainly used for patients with chronic palsy, published in detail elsewhere [31,32,33]. For this study, the surgical procedures were grouped into facial nerve reconstruction procedures, muscle transposition and static facial procedures, and eyelid surgery. The non-surgical treatments were categorized into physical therapy/speech therapy, electrotherapy, botulinum toxin injection, eye moisture chamber, eye drops, facial exercises at home, and in-house daycare facial biofeedback training.

### 2.5. Statistics

Statistical analyses were performed using IBM SPSS version 26.0 statistical software for Windows (Chicago, IL, USA). Maps to demonstrate the regional distributions of the referred patients were prepared with the software program PLZ-Diagramm 3.8 (Wessiepe, Grevenbroich, Germany). Differences between two independent subgroups for nominal data were compared with Pearson’s chi-square test and between more than two subgroups with Fisher’s exact test. Differences between two independent subgroups for metric data were compared with the *t*-test. Differences between two dependent subgroups for metric data were compared with the paired *t*-test and between more than two subgroups with univariate analysis of variance (ANOVA). A Bonferroni correction was applied in case of multiple comparisons in-between >2 subgroups. The probability of complete recovery from acute facial palsy was calculated by the Kaplan–Meier method. Recovery differences of the two subgroups were compared by the log-rank test. Multivariable analysis was performed using the Cox proportional hazards model to estimate the hazard ratio (HR) for recovery. For all statistical tests, significance was two-sided and set to *p* < 0.05.

## 3. Results

### 3.1. Baseline and Facial Palsy Characteristics

Aggregate baseline data on the entire study group of 1220 patients (58.4% female, median age: 50 years) are presented in Appendix A. The characteristics showed differences between patients referred with acute versus chronic facial palsy (Table 1). The patients with acute palsy were referred at a median time of 1 day (range: 0–90); 76.2% of the patients with acute palsy were referred within 72 h after onset. The median time for patients with idiopathic etiology was 2 days (range: 0–90); 72.1% of the patients with idiopathic etiology were referred within 72 h. The patients with chronic palsy were referred at a median time of 1.5 years (range: 0.2–70). The proportion of female patients in the chronic palsy group was higher than in the acute palsy group (*p* < 0.0001). The rate of recurrent palsies was higher in the acute palsy group (*p* = 0.039). Patients with tumors as underlying etiology and congenital cases presented more frequently as chronic cases (*p* < 0.0001). The patients with acute palsy were older than chronic palsy cases (*p* < 0.0001).

### 3.2. Diagnostics

The need for diagnostic tests was different in the acute and chronic palsy groups for all analyzed diagnostics (nearly all *p* < 0.001; Table 2; data for all patients together are in Appendix A). Imaging (neck sonography, magnetic resonance imaging, computed tomography) and serology were relevant for acute but not chronic cases. In contrast, since its introduction, facial sonography has been important for decision-making in chronic flaccid cases. Classical topography tests were predominately needed in acute cases. Standardized facial photo series were very important in all cases but even more important in chronic cases to document the changes during follow-up and to allow controlled facial grading. During follow-up, electrophysiological tests were by far the most frequently repeated investigation (Appendix A). The mean number of visits was 2.6 ± 3.6 (range: 1–52). If the patients were referred in the acute phase, the mean treatment time in the center was 0.9 ± 1.9 years. The treatment time for patients with chronic palsy was 1.4 ± 1.9 years.

### 3.3. Treatment

Drug treatment was the most important type of therapy for patients with acute facial palsy (Table 3). Glucocorticoid, acyclovir, the combination of glucocorticoid ± acyclovir, and antibiotics, if indicated, were started with a median time of 1–2 days after onset (Appendix A). Conservative eye care (eye moisture chamber, eye drops/ointment) as well as instructions for facial exercises and facial care at home were predominately used for patients with acute palsy. Upper eyelid weight surgery was important for patients with severe acute and chronic flaccid palsy. Other eyelid surgery was predominately reserved for chronic cases. Facial–facial nerve suture and facial nerve interpositional graft were important for acute cases with severe degenerative lesions without the possibility for spontaneous regeneration. Hypoglossal–facial-nerve jump suture was the predominant choice out of the nerve surgery techniques for chronic flaccid cases. Some patients had received surgery before referral to the center (*n* = 71, eyelid surgery; *n* = 26, facial nerve surgery; *n* = 11, sling surgery). Three patients were referred to other facial specialists (*n* = 1; Masseteric-facial nerve suture plus facial cross-face suture; *n* = 1; facial cross-face suture; *n* = 1, temporalis muscle mini-transfer). Electrotherapy, botulinum toxin injection, and facial EMG biofeedback training were important options in the chronic phase of the disease, the former procedure for chronic flaccid cases and the latter two procedures for patients with facial synkinesis.

### 3.4. Facial Nerve Function at Initial and Last Visits

Facial nerve function in patients with acute or chronic facial palsy was highly variable (Appendix A). The relation between incomplete and complete facial palsy in the acute phase was 5:1. Post-paralytic synkinesis was the dominant status of patients referred in the chronic phase. The number of visits to the center varied from 1 to 52. The average interval between initial and last presentation was 1.1 ± 1.9 years. At the initial visit, male patients had significantly worse facial function than female patients (*p* < 0.0001), but in all PROMs, they had better facial-specific (FDI, FaCE) and general (SF-36) quality of life (*p* < 0.0001 for nearly all domains; Table 4). Patients >50 years of age had a worse facial function and lower PROM values in most subdomains (*p* < 0.0001 for nearly all domains). Patients with chronic facial palsy (flaccid and synkinesis) had better facial function than patients with acute palsy but lower FDI, FaCE, and SF-36 values in most subdomains. Patients with facial palsy related to a tumor had the lowest quality of life values. At the last visit, the gender influence was smaller; female patients displayed a lower quality of life in fewer subdomains (Table 5). Patients >50 years of age and patients with chronic palsy still showed a lower quality of life in most subdomains.

### 3.5. Outcome

The interval between first and last presentation for all patients was 1.1 ± 1.9 years (acute palsy: 0.9 ± 1.9; chronic palsy: 1.4 ± 1.9 years). The outcome measures for the first and last visit for the different treatments are shown for the Stennert index and the FDI in Table 6 and for the FaCE and the SF-36 in Table 7. The absolute improvement of facial grading for selected subgroups is presented in Figure 1. Overall, facial grading improved highly significantly (*p* < 0.001) for patients with acute or chronic palsy during the treatment in the center. The changes of the PROMs are presented in Figure 2. Treatment with glucocorticoids, acyclovir, and antibiotics was followed by a highly significant improvement of all outcome measures (*p* < 0.001). The effect was less notable for the SF-36 (*p* < 0.05). In case of chronic flaccid palsy, hypoglossal–facial-nerve jump suture, any eyelid surgery, and especially an upper eyelid weight led to significant improvement of facial motor function (*p* < 0.001) and facial-specific quality of life (FDI, FaCE; *p* < 0.05), but not to an improvement in the general quality of life (SF-36; *p* > 0.05). Conservative measures, such as physical therapy, botulinum toxin injections, special facial exercises at home, or facial EMG feedback training, also led to highly significant improvements according to facial grading, FDI, and FaCE (mostly, *p* < 0.001). Again, the effect was not highly significant for the SF-36. There was one exception, general quality of life was significantly improved when using botulinum toxin injections for facial synkinesis (*p* < 0.001). In addition, Appendix A regards the improvement from the perspective of acute and chronic facial palsy separately and for the PROM subdomains. Treatment of patients with acute palsy improved in nearly all subdomains highly significantly (*p* < 0.001), for general quality of life significantly (SF-36; *p* < 0.05). Patients with chronic facial palsy showed highly significant improvements (mostly *p* < 0.001), except for some SF-36 subdomains (*p* > 0.05). Glucocorticoids alone for acute facial palsy were effective in regard to the improvement of facial grading, FDI, and FaCE, but not for the SF-36 subdomains (Appendix A). Botulinum toxin injections for facial synkinesis were also effective, seen in the improvement of facial grading and of FDI and in some subdomains of FaCE and SF-36 (mostly *p* < 0.001). The improvement for patients with facial synkinesis was even greater after EMG facial biofeedback training (mostly *p* < 0.001; Appendix A). The improvement was less pronounced for surgical procedures, analyzed for the larger subgroups of hypoglossal–facial jump surgery and eyelid surgery: Facial grading was significantly improved, along with some FDI subdomains and a few FaCE subdomains, and only the SF-36 physical functioning subdomain after hypoglossal–facial jump surgery.

Finally, the recovery rates after acute facial palsy were analyzed. Univariate analysis showed that idiopathic etiology, incomplete palsy, low initial Stennert grading, high initial FDI or FaCE, no pathological spontaneous activity in EMG, normal stapedial reflex, prednisolone, or combined prednisolone and acyclovir treatment were beneficial parameters associated to higher complete recovery rates (Appendix A, Appendix A; all *p* < 0.05). Multivariate modeling (Appendix A) showed that initial better facial grading (hazard ratio (HR) 1.571; confidence interval (CI) 1.026 to 2.404, *p* = 0.038) and FaCE total score (HR 2.653; CI 1.519 to 4.635; *p*= 0.001) predicted higher probability of complete recovery. Idiopathic etiology was related to better outcome (HR 1.320; CI 1.000 to 1.742, *p* = 0.050) and an iatrogenic lesion to worse outcome (HR 0.485; CI 0.354 to 0.665; *p* < 0.0001). Out of the diagnostic tests, a normal stapedial reflex remained an independent predictor of better outcome (HR 2.077; CI 1.498 to 2.88; *p* < 0.0001). Concerning acute therapy, prednisolone alone or in combination was related to better outcome (HR 3.614; CI 2.124 to 6.149; *p* < 0.0001).

## 4. Discussion

Data on the outcome of treatment for acute or chronic facial palsy based on large series from multidisciplinary facial nerve centers are sparse. Specifically, large series reporting standardized outcome measures of a treatment in large-volume facial nerve centers are lacking. To our knowledge, the series from the facial nerve center in the Massachusetts Eye and Ear Infirmary, Boston, with 1989 patients treated between 2003 and 2013, is, so far, the largest [4]. The series from Boston presents patient and treatment characteristics as well as algorithms for decision-making. Outcome measures are not presented in this series. This also applies to the recent series of the facial nerve unit in Madrid [5]. The recent series from the Sydney Facial Nerve Clinic is much smaller, analyzing 145 patients treated between 2015 and 2018 [10], but it reports, at least, facial grading and facial-specific PROM data at initial referral. Krane et al. from Oregon focused on the surgical reanimation of 22 patients when reporting about the implementation of a facial nerve center with initial data from 2014 and 2019 [9]. They point out how important it is to measure outcomes when using a multidisciplinary approach to build up a facial nerve center. They used quantitative data (FACE-gram and Emotrics) as well as PROMs (FaCE and the Synkinesis Assessment Questionnaire (SAQ)) [34,35,36]. Moreover, a recent meta-analysis has shown that facial grading only explains a small part of the patient’s quality of life; the association between grading and facial PROMs is low to moderate [37]. The present study is the first using a large and unselected series of a facial nerve center, beyond facial grading, with initial and follow-up FaCE, FDI, and SF-36 data. Confirming the results of other studies, facial-specific quality of life data (FaCE, FDI) is much more informative and meaningful than general quality of life data (SF-36) [38].

The number of studies measuring the outcome of a treatment for acute facial palsy with PROMs is small. Even the landmark phase III clinical trials of prednisolone treatment for Bell’s palsy primarily relied on subjective facial grading [39,40]. In a small but prospective series of 21 patients with Bell’s palsy, all FaCE subdomains significantly improved after corticosteroid treatment [41]. In a recent Swedish prospective trial of 96 patients with Bell’s palsy, both the FDI and FaCE improved over time after corticosteroid treatment [42]. In these two studies, again, the correlation between facial grading and the PROM results was low to moderate. The present study could confirm these results in a large cohort of patients with Bell’s palsy for a routine clinical setting in a specialized center but beyond clinical trials. PROM data for non-idiopathic acute palsy are given here for the first time.

The situation is very different for patients receiving facial nerve reconstruction surgery. Specifically, the facial nerve center in Boston (but also others) has published the outcome for selected cases, usually focused on a specific technique of reconstruction, based on PROMs such as the FDI and FaCE [33,43,44,45,46,47,48]. As in the present study, the effects of surgery are much better presented by PROM measures rather than facial grading. The same holds true for the treatment of synkinesis by, for instance, botulinum toxin treatment or other conservative measures [49,50].

The present study has several limitations. The retrospective analysis can suggest associations but cannot analyze any causality. Although decision-making for diagnostics and treatment follows general rules in the center, the individual decision or deviation from standards could not be analyzed. All patients were treated. Hence, we cannot report on the spontaneous improvement or deterioration of the disease. In contrast to prospective trials, the facial grading and PROM measurements were not performed at defined follow-up times. To overcome this limitation, the Kaplan–Meier method was used to include time as a variable when measuring the outcome after treatment of acute facial palsy. Concerning outcomes for the treatment of chronic facial palsy, a sufficient follow-up time was ensured: 74% and 64% of the chronic facial palsy group had a follow-up of longer than 4 and 6 months, respectively. As is typical for the treatment of chronic facial palsy, most patients received combinations of different treatments (several types of surgery, combination of surgical with non-surgical treatment). Therefore, it was not feasible to perform a multivariable analysis after the univariable analysis (focus on treatment). The subgroups of identical treatment combinations were too small to allow multivariable analysis. Therefore, we could not analyze the effect of the interaction between the different treatment types on the outcome in patients with chronic facial palsy. With 95 cases, children were a minority; hence, they may be underrepresented. Patients with congenital lesions and candidates for free flap reconstruction were underrepresented as well. The evolution within the last few years show that the referral of such cases will increase in the future. Other facial nerve centers, including ours, have focused on the standardization of diagnostics, treatment, and outcome measures [4,35]. The next step will be the introduction of objective, automated, therapist-independent measurement tools to evaluate the outcome. After a decade of research, the first automated tools feasible for use in clinical routine have been published [51,52]. Furthermore, the referral to comprehensive facial palsy services still needs to be improved by the continuous training of our colleagues. The rate of treated cases of Bell’s palsy without recovery, referred to secondary care, still seems to be too low; hence, many patients with synkinetic recovery remain untreated [2]. In the present study, 44% and 39% of the chronic cases were referred not before 12 and 9 months, respectively, after onset.

## 5. Conclusions

In this large and unselected series of patients with acute facial palsy, chronic flaccid paralysis, or post-paralytic synkinesis, it has been shown that a standardized approach in a multidisciplinary and inter-professional facial nerve center leads to the improvement of facial function and the related quality of life. To reach satisfactory results, a fast referral of complex acute cases, if recovery of normal facial function is unlikely or definitive, is recommended. Timely referral is very important. The center has to guarantee a large spectrum of up-to-date diagnostics, surgical techniques, and conservative measures to decide the best solution in individual cases.

## Figures and Tables

**Figure 1 jcm-11-00427-f001:**
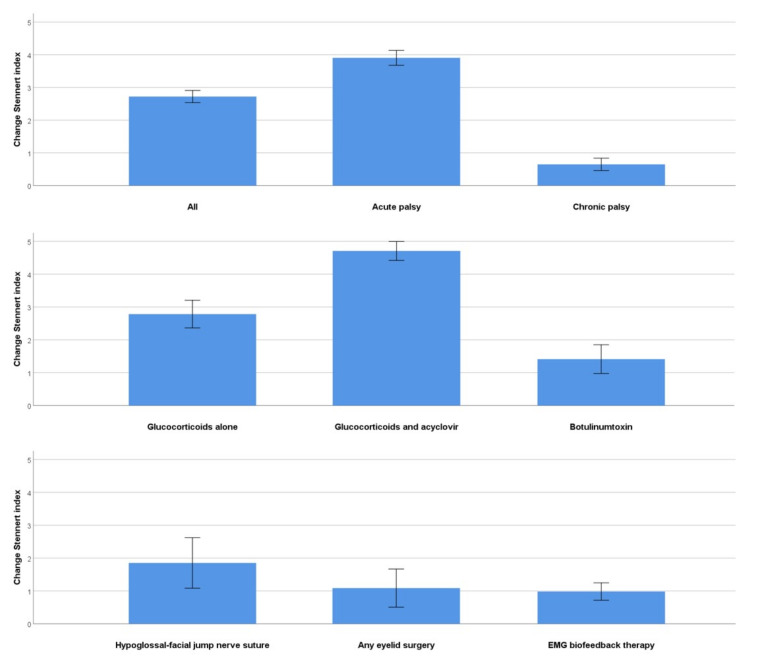
Absolute improvement of facial function between initial and last evaluation in the Facial Nerve Center, assessed with the Stennert index (higher number = more improvement; absolute total Stennert index ranges from 0 to 10). The overall improvement of the complete study for the subgroup of acute versus chronic facial palsy as well as of several important subgroups with different therapy scenarios is shown.

**Figure 2 jcm-11-00427-f002:**
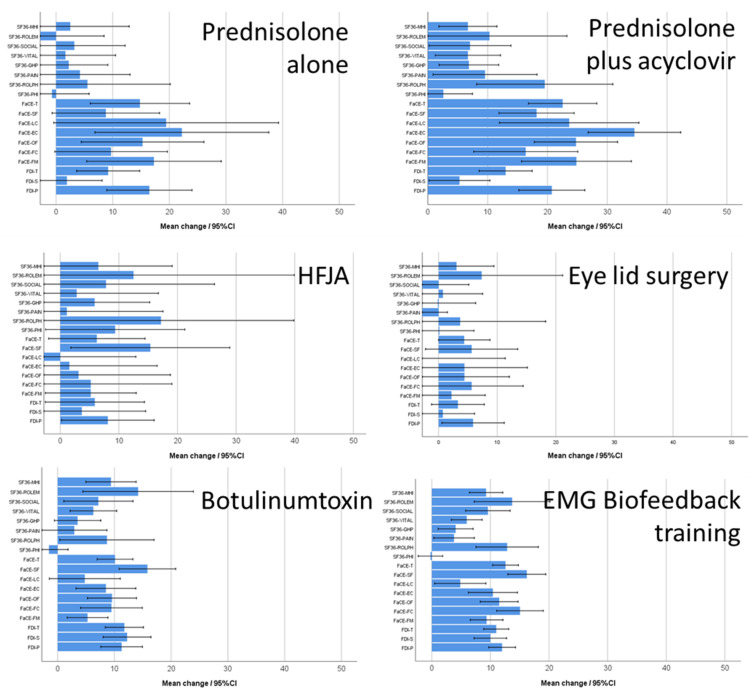
Absolute improvement of facial function between initial and last evaluation in the Facial Nerve Center, assessed with the Facial Disability Index (FDI), the Facial Clinimetric Evaluation Scale (FaCE), and 36-Item Short Form Survey (SF-36) separately for the same subgroups as in Figure 1. FDI-P = FDI physical function, FDI-S = FDI social function; FDI-T = FDI total; FaCE-FM = FaCE facial movement; FaCE-FC = FaCE facial comfort; FaCE-OF = FaCE oral function; FaCE-EC = FaCE eye comfort; FaCE-LC = FaCE lacrimal control; FaCE-SF = FaCE social function; FaCE-T = FaCE total score; SF36-PHI = = SF-36 physical functioning; SF36-ROLPH = SF-36 physical role functioning; SF36-PAIN = SF-36 bodily pain; SF36-GHP = general health perceptions; SF36-VITAL = SF-36 vitality; SF36-SOCIAL = SF-36 social role functioning; SF36-ROLEM = SF-36 emotional role functioning; SF36-MHI = SF-36 mental health. More details on other treatment types are shown in Appendix A.

**Table 1 jcm-11-00427-t001:** Comparison of patients’ characteristics referred as a case of acute facial palsy (≤90 days after onset) versus as chronic facial palsy (>90 days after onset).

	Acute Palsy *n* = 697	Chronic Palsy *n* = 523		
**Parameter**	**Absolute (N)**	**Absolute (N)**	**Χ^2^, df**	** *p* ** *****
Gender				
Female	365	348	24.7, 1	**<0.0001**
Male	332	175		
Side			7.7, 2	**0.021**
Right	343	224		
Left	352	293		
Bilateral	2	6		
Recurrent palsy			4.5, 1	**0.039**
No	646	500		
Yes	51	23		
Localization			1.4, 2	0.490
peripheral	692	519		
central	5	3		
nuclear	0	1		
Etiology			69.3, 6	**<0.0001**
idiopathic	306	207		
Iatrogenic **	197	139		
infectious/inflammatory	104	94		
traumatic	19	24		
neoplastic, benign	11	31		
neoplastic, malignant	0	3		
congenital	0	25		
	**Mean ± SD**	**Mean ± SD**	**T, df**	** *p* **
Age (years) at initial diagnosis	53.0 ± 18.6	41.2 ± 20.3	10.6, 1218	**<0.0001**
Age (years) at initial presentation	53.0 ± 18.6	46.4 ± 18.2	6.2, 1218	**<0.0001**
Interval (years) onset to initial presentation	0.02 ± 0.04	5.3 ± 9.2	−15.3, 1218	**<0.0001**

* Significant *p*-values (*p* < 0.05) in bold for nominal data due to Pearson’s chi-square test for two subgroups or due to Fisher’s exact test for more than two subgroups and for metric data due to *t*-test; ** post-operative or post-radiotherapy; SD = standard.

**Table 2 jcm-11-00427-t002:** Comparison of necessary diagnostics in the case of acute facial palsy (≤90 days after onset) versus chronic facial palsy (>90 days after onset).

	Acute Palsy *n* = 697	Chronic Palsy *n* = 523		
**Parameter**	**Absolute (N)**	**Absolute (N)**	**Χ^2^, df**	** *p* ** *****
Sonography of the neck			364.2, 1	**<0.0001**
Yes	451	54		
No	246	469		
Sonography of the facial muscles			51.5, 1	**<0.0001**
Yes	65	128		
No	632	395		
Magnet resonance imaging, cranial			38.2, 1	**<0.0001**
Yes	112	25		
No	585	498		
Computed tomography, cranial			44.2, 1	**<0.0001**
Yes	80	8		
No	617	515		
Facial electrophysiology			5.1, 1	**0.024**
Yes	596	470		
No	101	53		
Facial photo series			31.6, 1	**<0.0001**
Yes	613	503		
No	79	14		
Audiometry			10.9, 1	**0.001**
Yes	508	335		
No	189	188		
Tympanometry			18.3, 1	**<0.0001**
Yes	496	311		
No	201	212		
Stapedius reflex test			14.3, 1	**<0.0001**
Yes	482	307		
No	215	216		
Gustatory test			47.3, 1	**<0.0001**
Yes	477	256		
No	220	267		
Vestibular tests			101.9, 1	**<0.0001**
Yes	484	212		
No	213	311		
Schirmer test			47.1, 1	**<0.0001**
Yes	488	233		
No	249	290		
Serology			454.5, 1	**<0.0001**
Yes	429	12		
No	268	511		

* Significant *p*-values (*p* < 0.05) in bold; Pearson’s chi-square test.

**Table 3 jcm-11-00427-t003:** Comparison of treatment in the case of acute facial palsy (≤90 days after onset) versus chronic facial palsy (>90 days after onset).

	Acute Palsy *n* = 697	Chronic Palsy *n* = 523		
**Parameter**	**Absolute (N)**	**Absolute (N)**	**Χ^2^, df**	** *p* ** *****
Glucocorticoids			760.0, 1	**<0.0001**
Yes	564	7		
No	133	516		
Acyclovir			413.5, 1	**<0.0001**
Yes	384	2		
No	313	521		
Antibiotics			39.9, 1	**<0.0001**
Yes	60	3		
No	637	520		
Facial nerve reconstruction				
Facial nerve reconstruction, any	16	33	14.7, 1	**0.001**
Facial-facial nerve suture	2	2	0.01, 1	1.000
Facial nerve interpositional graft	6	2	1.1, 1	0.478
Hypoglossal–facial-nerve jump suture	10	31	18.6, 1	**<0.0001**
Muscle and sling plasty				
Temporal muscle transfer	0	1	1.3, 1	0.429
Sling plasty angle of the mouth	4	12	6–8, 1	**0.011**
Eyelid surgery				
Upper eyelid weight	31	46	9.6, 1	**0.003**
Tarsorrhaphy	1	9	13.2, 1	**0.001**
Kanthopexy	3	9	8.4, 1	**0.006**
Brow plasty	7	15	10.8, 1	**0.002**
Blepharoplasty	2	13	17.5, 1	**0.001**
Lower lid plasty	8	17	12.1, 1	**0.013**
Non-surgical adjuvant therapy				
Physical therapy/speech therapy	55	47	0.5, 1	0.531
Electrotherapy	11	47	51.5, 1	**<0.0001**
Botulinumtoxin injection	41	132	92.0, 1	**<0.0001**
Eye moisture chamber	423	12	444.1, 1	**<0.0001**
Eye drops/ointment	430	20	430.0, 1	**<0.0001**
Facial exercises at home	441	14	469.1, 1	**<0.0001**
Facial EMG biofeedback training	31	255	326.9, 1	**<0.0001**

* Significant *p*-values (*p* < 0.05) in bold; Pearson’s chi-square test.

**Table 4 jcm-11-00427-t004:** Initial facial function of patients by demographics and etiology.

Parameter	SI, Total (Mean ± SD)	FDI, Physical (Mean ± SD)	FDI, Social (Mean ± SD)	FDI, Total (Mean ± SD)	FaCE, Total (Mean ± SD)	SF-36, PFI (Mean ± SD)	SF-36, GHP (Mean ± SD)
Gender							
Female	4.5 ± 3.0	64.2 ± 20.4	66.2 ± 21.1	65.1 ± 18.0	57.3 ± 22.8	80.3 ± 26.4	57.5 ± 21.0
Male	5.1 ± 3.0	69.7 ± 20.1	74.5 ± 20.4	72.0 ± 17.2	67.7 ± 21.7	85.0 ± 21.6	62.8 ± 20.8
T, df	−3.6, 1218	−4.0, 846	−5.9, 841	−5.9, 847	−6.4, 829	−2.8, 811	−3.5, 796
*p* *	**<0.0001**	**<0.0001**	**<0.0001**	**<0.0001**	**<0.0001**	**0.005**	**<0.0001**
Age at first presentation							
< median 50 years	4.4 ± 2.9	69.6 ± 20.0	70.5 ± 20.6	70.0 ± 17.9	62.7 ± 22.1	88.9 ± 20.3	61.4 ± 21.4
≥ median 50 years	5.0 ± 3.1	63.1 ± 20.4	68.5 ± 20.9	65.5 ± 17.9	60.8 ± 23.9	72.6 ± 25.4	58.2 ± 19.9
T, df	−3.6, 1218	4.7, 846	1.8, 841	3.7, 847	1.6, 829	8.5, 811	4.4, 796
*p* *	**<0.0001**	**<0.0001**	0.065	**<0.0001**	0.118	**<0.0001**	**<0.0001**
Interval onset to first presentation							
≤90 days (acute facial palsy)	5.1 ± 2.9	69.6 ± 21.1	75.9 ± 18.1	72.6 ± 17.3	72.3 ± 23.2	80.2 ± 24.3	59.8 ± 20.1
>90 days (chronic facial palsy)	4.2 ± 3.1	63.7 ± 19.5	64.2 ± 21.2	63.9 ± 17.6	53.8 ± 19.2	80.8 ± 24.7	58.8 ± 22.2
T, df	5.3, 1218	4.0, 846	8.0, 841	7.0, 847	12.5, 829	−1.1, 811	2.3, 796
*p* *	**<0.0001**	**<0.0001**	**<0.0001**	**<0.0001**	**<0.0001**	0.253	**0.023**
Etiology							
idiopathic	4.7 ± 2.8	68.7 ± 20.8	70.4 ± 21.3	69.3 ± 18.4	65.5 ± 22.8	83.2 ± 22.7	59.8 ± 20.9
iatrogenic	4.4 ± 3.1	64.6 ± 18.3	70.2 ± 19.9	67.4 ± 16.5	59.4 ± 21.5	78.4 ± 24.5	61.7 ± 20.2
infectious/inflammatory	4.8 ± 2.9	60.3 ± 21.4	64.7 ± 20.2	62.6 ± 18.8	57.3 ± 25.2	74.4 ± 28.1	55.3 ± 23.7
traumatic	4.6 ± 3.2	72.1 ± 21.8	70.0 ± 20.9	70.8 ± 17.5	60.8 ± 22.4	76.4 ± 31.6	54.9 ± 18.5
neoplastic, benign	7.9 ± 2.4	62.7 ± 14.7	64.8 ± 16.7	63.8 ± 12.4	47.2 ± 12.5	83.5 ± 17.4	56.5 ± 17.2
neoplastic, malignant	8.3 ± 1.2	60.0 ± 0	82.0 ± 8.5	71.0 ± 4.2	45.8 ± 20.0	90.0 ± 14.4	61.0 ± 1.4
congenital	4.9 ± 2.7	87.5 ± 13.2	86.8 ± 13.7	87.1 ± 11.2	72.8 ± 16.2	100.0 ± 0	75.8 ± 24.6
F, df	9.5, 6	7.0, 6	3.4, 6	6.6, 6	6.5, 6	4.2, 6	3.8, 6
*p* *	**<0.0001**	**0.003**	**0.001**	**<0.0001**	**<0.0001**	**<0.0001**	**0.001**

SD = standard deviation; SI = Stennert index; FDI = Facial Disability index; FaCE = Facial Clinimetric Evaluation Scale; SF-36 = 36-Item Short Form Survey; PF = physical function subdomain; GH = general health subdomain. * Significant *p*-values (*p* < 0.05) in bold; *t*-test for comparison of two subgroups; univariate ANOVA for more than two subgroups.

**Table 5 jcm-11-00427-t005:** Last facial function of patients by demographics and etiology.

Parameter	SI, Total (Mean ± SD)	FDI, Physical (Mean ± SD)	FDI, Social (Mean ± SD)	FDI, Total (Mean ± SD)	FaCE, Total (Mean ± SD)	SF-36, PFI (Mean ± SD)	SF-36, GHP (Mean ± SD)
Gender							
Female	2.0 ± 2.5	74.7 ± 17.4	74.7 ± 17.0	74.7 ± 15.2	64.7 ± 20.1	83.4 ± 22.6	63.3 ± 21.2
Male	2.2 ± 2.9	78.0 ± 17.6	79.0 ± 17.8	84.4 ± 26.5	73.2 ± 19.7	82.1 ± 25.7	61.6 ± 20.4
T, df	−0.6, 1072	−1.6, 621	−2.6, 617	−2.5, 617	−4.0, 618	−0.2, 618	−0.1, 618
*p* *	0.519	0.105	**0.011**	**0.014**	**<0.0001**	0.781	0.679
Age at first presentation							
< median 50 years	2.3 ± 2.7	79.3 ± 16.3	76.7 ± 17.2	78.1 ± 15.1	69.1 ± 19.4	90.8 ± 17.6	65.5 ± 21.2
> median 50 years	1.9 ± 2.7	71.8 ± 18.0	75.5 ± 17.6	73.7 ± 56.8	65.9 ± 21.2	74.9 ± 26.4	59.6 ± 20.2
T, df	2.5, 1072	4.4, 621	0.6, 617	2.6, 617	1.1, 618	6.4, 618	3.7, 618
*p* *	**0.012**	**<0.0001**	0.499	**0.009**	0.295	**<0.0001**	**<0.0001**
Interval onset to first presentation							
≤90 days (acute facial palsy)	1.2 ± 2.3	80.1 ± 18.1	80.1 ± 16.1	85.4 ± 23.1	77.6 ± 21.1	84.0 ± 22.6	64.2 ± 25.7
>90 days (chronic facial palsy)	3.6 ± 2.7	73.3 ± 16.7	73.8 ± 17.7	73.6 ± 15.2	62.7 ± 17.9	81.1 ± 25.7	60.2 ± 19.5
T, df	−15.2, 1072	3.8, 621	2.4, 617	3.6, 617	8.2, 618	−0.7, 618	0.2, 618
*p* *	**<0.0001**	**<0.0001**	**0.017**	**<0.0001**	**<0.0001**	0.533	0.812
Etiology							
idiopathic	1.4 ± 2.0	77.6 ± 17.4	77.9 ± 16.4	77.8 ± 14.7	72.9 ± 18.6	86.3 ± 21.6	66.7 ± 22.4
iatrogenic	2.5 ± 2.9	75.1 ± 17.4	76.2 ± 16.9	82.4 ± 17.3	62.7 ± 20.9	83.2 ± 19.9	58.2 ± 14.9
infectious/inflammatory	2.0 ± 2.3	74.0 ± 16.8	73.7 ± 19.7	77.9 ± 14.4	65.9 ± 20.1	70.5 ± 14.0	54.8 ± 24.6
traumatic	3.0 ± 3.2	82.2 ± 18.1	77.5 ± 13.5	79.8 ± 13.6	73.3 ± 17.9	92.0 ± 17.9	77.0 ± 15.4
neoplastic, benign	7.4 ± 2.0	60.9 ± 14.8	65.2 ± 18.3	63.0 ± 14.3	49.5 ± 17.6	84.2 ± 22.2	65.8 ± 15.0
neoplastic, malignant	10 ± 0	75.0 ± 0	80.0 ± 0	77.5 ± 0	48.3 ± 0	NA	NA
congenital	4.2 ± 2.8	96.3 ± 2.5	92.8 ± 3.6	94.5 ± 2.0	83.3 ± 4.1	95.0 ± 0	62.0 ± 0
F, df	42.4, 6	3.3, 6	1.7, 6	3.0, 6	5.1, 6	2.0, 6	2.8, 6
*p* *	**<0.0001**	**0.003**	0.108	0.008	**<0.0001**	0.066	**0.010**

SD = standard deviation; SI = Stennert index; FDI = Facial Disability index; FaCE = Facial Clinimetric Evaluation Scale; SF-36 = 36-Item Short Form Survey; PFI = physical function subdomain; GHP = general health subdomain; * significant *p*-values (*p* < 0.05) in bold; *t*-test for comparison of two subgroups; univariate ANOVA for more than two subgroups.

**Table 6 jcm-11-00427-t006:** Comparison of outcome measures using the Stennert index (sum) and the Facial Disability index (total) between first and last visits *.

		Stennert Index, Sum				Facial Disability Index, Total		
	First Visit Mean ± SD	Last Visit Mean ± SD	T, df	*p*	First Visit Mean ± SD	Last Visit Mean ± SD	T, df	*p* **
All	4.8 ± 3.0	2.1 ± 2.7	28.8, 1073	**<0.0001**	64.8 ± 16.6	76.0 ± 16.7	−14.7, 604	**<0.0001**
Glucocorticoids	5.1 ± 2.7	1.1 ± 2.1	32.1, 570	**<0.0001**	68.3 ± 6.9	87.4 ± 70.1	−82, 327	**0.006**
Acyclovir	5.6 ± 2.5	1.0 ± 2.0	32.1, 384	**<0.0001**	68.7 ± 16.4	81.0 ± 15.0	−5.2, 170	**<0.001**
Glucocorticoids± Acyclovir	5.7 ± 2.5	1.0 ± 2.0	32.8, 380	**<0.0001**	62.5 ± 19.4	80.5 ± 17.6	−6.0, 180	**<0.0001**
Antibiotics	5.6 ± 2.7	1.8 ± 2.9	10.0, 62	**<0.0001**	65.9 ± 12.6	78.9 ± 27.4	−3.8, 18	**0.021**
Facial nerve reconstruction, any	8.6 ± 1.7	6.8 ± 2.6	5.1, 48	**<0.0001**	67.3 ± 16.7	71.3 ± 19.9	−2.4, 32	**0.019**
Facial–facial nerve suture	NA	NA	NA	NA	74.25 ± 7.4	72.7.5 ± 8.7	3.0, 1	0.205
Facial nerve interpositional graft	8.3 ± 1.5	5.9 ± 3.0	2.1, 7	0.074	73.5 ± 7.4	73.5 ± 8.7	0, 3	1.000
Hypoglossal–facial-nerve jump suture	8.7 ± 1.3	6.8 ± 2.5	4.9, 40	**<0.0001**	66.2 ± 16.2	77.5 ± 20.0	−2.4, 28	**0.020**
Sling plasty angle of the mouth	8.2 ± 2.2	9.3 ± 1.3	−2.5, 15	**0.023**	62.8 ± 14.3	52.4 ± 10.7	21.2, 13	0.056
Eyelid surgery, any	7.2 ± 2.9	6.1 ± 3.1	3.7, 91	**<0.0001**	61.6 ± 17.1	64.0 ± 19.4	−2.1, 52	**0.042**
Upper eyelid weight	7.7 ± 2.5	6.4 ± 3.0	4.1, 76	**<0.0001**	61.2 ± 17.9	63.2 ± 20.6	−1.8, 42	0.072
Tarsorrhaphy	7.0 ± 2.5	5.9 ± 2.8	0.9, 8	0.384	59.0 ± 14.7	62.1 ± 16.3	−0.6, 7	0.550
Kanthopexy	7.8 ± 2.5	8.2 ± 2.3	−0.4, 11	0.701	63.0 ± 13.7	42.3 ± 21.0	1.8, 5	0.135
Brow plasty	7.2 ± 3.1	6.7 ± 3.6	0.8, 21	0.451	67.1 ± 15.4	60.5 ± 23.8	0.8, 12	0.427
Blepharoplasty	4.4 ± 3.6	4.5 ± 2.8	−0.1, 14	0.896	57.3 ± 17.3	65.3 ± 16.6	−2.6, 12	**0.025**
Lower lid plasty	8.6 ± 1.9	7.8 ± 2.3	1.7, 24	0.100	59.5 ± 23.2	57.7 ± 25.2	−1.4, 12	0.811
Physical therapy/speech therapy	5.9 ± 3.0	3.3 ± 3.0	7.9, 96	**<0.0001**	62.5 ± 14.9	70.3 ± 19.4	−5.4, 65	**0.002**
Electrotherapy	7.3 ± 2.3	5.9 ± 2.9	4.1, 52	**<0.0001**	62.1 ± 15.9	70.8 ± 17.1	3.9, 46	**<0.0001**
Botulinumtoxin injection	3.7 ± 2.9	2.3 ± 1.9	6.4, 159	**<0.0001**	60.6 ± 15.9	72.0 ± 15.7	−6.7, 110	**<0.0001**
Eye moisture chamber	6.0 ± 2.4	1.3 ± 2.4	32.6, 434	**<0.0001**	65.0 ± 17.8	65.0 ± 17.8	−2.6, 87	**0.010**
Eye drops/ointment	5.9 ± 2.5	1.3 ± 2.4	32.1, 448	**<0.0001**	65.7 ± 16.6	78.7 ± 16.6	−7.7, 112	**<0.0001**
Facial exercises at home	5.4 ± 2.6	1.0 ± 2.0	31.8, 452	**<0.0001**	68.7 ± 16.1	80.8 ± 16.6	−5.5, 85	**<0.0001**
Facial EMG biofeedback training	3.8 ± 2.8	2,8 ± 2.0	7.3, 284	**<0.0001**	63.1 ± 1.2	74.3 ± 15.3	−10.2, 195	**<0.0001**

* *n* = 1073 patients with at least two visits; ** significant *p*-values (*p* < 0.05) in bold; paired *t*-test; NA = not applicable; SD = standard deviation.

**Table 7 jcm-11-00427-t007:** Comparison of outcome measures using FaCE (total score) and SF-36 (general health subdomain) between first and last visits *.

		FaCE, Total Score				SF-36, General Health		
	First Visit Mean ± SD	Last Visit Mean ± SD	T, df	*p*	First Visit Mean ± SD	Last Visit Mean ± SD	T, df	*p* **
All	53.8 ± 20.0	67.3 ± 20.8	−15.8, 613	**<0.0001**	58.4 ± 19.6	61.9 ± 21.7	−3.7, 599	**<0.0001**
Glucocorticoids	60.1 ± 21.5	80.8 ± 20.0	−10.2, 322	**<0.0001**	59.0 ± 16.7	65.0 ± 21.8	−2.3, 309	**0.014**
Acyclovir	62.8 ± 19.4	83.8 ± 16.9	−8.0, 158	**<0.0001**	58.2 ± 17.5	65.0 ± 23.4	−2.3, 140	**0.028**
Glucocorticoids± Acyclovir	60.5 ± 20.8	82.6 ± 17.9	−8.7, 177	**<0.0001**	59.6 ± 17.5	64.4 ± 22.9	−2.0, 171	**0.047**
Antibiotics	53.2 ± 20.5	80.7 ± 27.8	−6.9, 17	**<0.0001**	54.4 ± 24.1	65.4 ± 19.7	−0.6, 14	0.066
Facial nerve reconstruction, any	54.9 ± 19.3	60.6 ± 18.7	−3.1, 32	**0.005**	59.0 ± 24.3	61.2 ± 13.5	−0.5, 26	0.601
Facial–facial nerve suture	70.8 ± 18.5	68.1 ± 15.9	0.4, 1	0.783	59.5 ± 10.4	53.5 ± 15.6	6.0, 1	0.105
Facial nerve interpositional graft	51.7 ± 11.7	53.0 ± 8.4	−0.1, 3	0.917	64.6 ± 10.4	61.5 ± 15.6	0.8, 3	0.469
Hypoglossal–facial-nerve jump suture	51.0 ± 18.5	59.3 ± 18.2	−3.1, 28	**0.005**	58.6 ± 26.9	61.2 ± 14.9	−0.7, 22	0.469
Sling plasty angle of the mouth	30.0 ± 4.7	29.2 ± 8.3	0.1, 2	0.942	42.5 ± 10.6	48.5 ± 12.0	−0.4, 4	0.772
Eyelid surgery, any	46.9 ± 18.1	50.7 ± 21.1	−2.5, 51	**0.015**	56.7 ± 23.9	54.2 ± 18.3	0.9, 47	0.390
Upper eyelid weight	48.1 ± 19.8	50.6 ± 23.1	−2.0, 42	**0.048**	54.8 ± 25.4	53.8 ± 17.8	0.4, 37	0.671
Tarsorrhaphy	41.7 ± 15.0	49.7 ± 14.3	−4.7, 6	**0.009**	48.5 ± 19.1	53.5 ± 5.0	−0.3, 6	0.818
Kanthopexy	44.7 ± 17.3	39.4 ± 17.2	0.9, 5	0.422	48.5 ± 2.1	40.0 ± 0	5.7, 5	0.111
Brow plasty	53.0 ± 15.5	55.2 ± 20.1	−0.4, 9	0.675	54.7 ± 10.8	50.7 ± 18.5	0.9, 7	0.475
Blepharoplasty	44.9 ± 17.9	49.8 ± 23.2	−1.1, 11	0.300	57.4 ± 14.1	53.0 ± 19.1	0.8, 12	0.462
Lower lid plasty	53.0 ± 21.3	55.0 ± 25.0	−0.4, 12	0.630	54.0 ± 36.2	47.3 ± 31.6	1.4, 9	0.184
Physical therapy/speech therapy	52.1 ± 20.3	62.5 ± 20.3	−5.9, 61	**<0.0001**	54.7 ± 14.7	55.8 ± 12.5	−0.2, 56	0.586
Electrotherapy	45.5 ± 18.1	54.4 ± 20.6	−4.7, 45	**<0.0001**	56.0 ± 18.8	57.1 ± 21.8	−0.4, 44	0.715
Botulinumtoxin injection	50.5 ± 19.7	61.1 ± 17.1	−6.5, 110	**<0.0001**	53.7 ± 20.2	62.1 ± 23.4	−1.9, 84	**<0.0001**
Eye moisture chamber	57.4 ± 22.2	78.7 ± 22.2	−9.2, 73	**<0.0001**	58.5 ± 18.3	66.5 ± 22.8	−2.6, 69	**0.013**
Eye drops/ointment	57.2 ± 21.4	77.6 ± 22.3	−10.0, 108	**<0.0001**	59.4 ± 17.4	66.6 ± 22.9	−1.9, 107	**0.020**
Facial exercises at home	63.3 ± 18.5	81.6 ± 18.5	−7.4, 72	**<0.0001**	59.5 ± 16.8	64.7 ± 22.8	−1.8, 60	0.076
Facial EMG biofeedback training	52.0 ± 17.9	63.9 ± 16.6	−10.7, 190	**<0.0001**	61.2 ± 21.6	61.5 ± 19.2	−0.9, 127	0.932

* *n* = 1073 patients with at least two visits; ** significant *p*-values (*p* < 0.05) in bold; paired *t*-test; FaCE = Facial Clinimetric Evaluation Scale; SF-36 = 36-Item Short Form Survey; NA = not applicable; SD = standard deviation.

## Data Availability

The datasets used during the current study are available from the corresponding.

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
