# Peer review of "Multidisciplinary Care of Patients with Facial Palsy: Treatment of 1220 Patients in a German Facial Nerve Center"

_jcm, 2022, doi:10.3390/jcm11020427_

Round 1

Reviewer 1 Report

This is an interested and important paper. It represents an important addition to the (scarce) literature on this topic. 

That said, the paper suffers from some potential problems related to the use of statistics with their data. 

1)

To me, the most important part of this paper is section 3.5, where authors present the influence of treatment for every outcome. However, based on Table 3, some patients received more than one treatment. How was the influence of multiple treatments handled? 

Did the authors select a subgrup of patient that received a single treatment? Or was everyone included regardless if they received one or multiple treatments? 

I believe that the first option should be used, and authors need to clearly state how many samples they use to compute their statistics. If a treatment cannot be isolated, then it is not easy to evalute its effect.

2) Figure 2 is too dificutl to follow. Please find another way to present this information, or select the most importnat points of the figure and leave the current figure as a supplement. 

3) The last part of section 3.5 is just an afterthough at this point and should be removed from the paper or significantly modified (lines 306-319)

4) There are a lof of very long sentences that make the paper difficult to follow. Please proofread your document to clarify the text. 

Some examples are:

  • (lines 60-64) As the patients are not only monitored by facial nerve grading but 62 from the beginning also by facial neve specific patient-related outcome measures 63 (PROMs), we will also present standardized outcome results.
  • (lines 42-45) Although several clinical guidelines based on high evidence clinical trials have been published, still many acute cases do not receive any or optimal treatment.  Moreover, patients with low probability to recover or already occurred incomplete recovery are not referred or referred very late to specialized facial palsy services [2,6].

Author Response

Point-by-point response

Manuscript ID: jcm-1509047

Thank you very much for the detailed comments of the three reviewers. We answer all queries point-by-point.

Reviewer #1

This is an interested and important paper. It represents an important addition to the (scarce) literature on this topic. 

That said, the paper suffers from some potential problems related to the use of statistics with their data. 

1) To me, the most important part of this paper is section 3.5, where authors present the influence of treatment for every outcome. However, based on Table 3, some patients received more than one treatment. How was the influence of multiple treatments handled? 

Did the authors select a subgroup of patient that received a single treatment? Or was everyone included regardless if they received one or multiple treatments? 

I believe that the first option should be used, and authors need to clearly state how many samples they use to compute their statistics. If a treatment cannot be isolated, then it is not easy to evaluate its effect.

Answer #2.1: The primary aim of this study was to show the efficacy of a multidisciplinary approach in facial nerve centers and to show from this perspective that the outcome can be satisfactory for treatment with acute as well as for patients with chronic facial palsy. The aim was not to analyze one special type of treatment it is often done in hospital-based studies (for instance, presenting results of one surgical technique to repair the facial nerve).

Therefore, we would like to answer this query separately for both patient groups, i.e. acute and chronic palsy, as the analyses were performed differently.

Acute palsy: It was possible that a patients with acute palsy got glucocorticoids, acyclovir or both as treatment. Hence, a combination was possible. This was also clearly mentioned already in the Abstract and was explained in the Methods at 2.4. The results for the combined use were already shown in Figure 2. We added now also in Table 4 and 5 the parameter “Glucocorticoids± Acyclovir”, showing the results for the combination. Because the patients were not analyzed at fixed time points during follow-up and different duration of follow-up, we used, as it is standard, the Kaplan-Meier method to include the effect of time, i.e. the variable follow-up time in-between the patients.

As it is a common standard (see Methods, 2.5 statistics), we first analyzed the association of single parameters on the outcome with the Kaplan-Meier method and log-rank test (univariable analysis, see also Discussion lines 390 ff). Significant parameters were then included into multivariable Cox models. Hence, we classically analyzed the interaction between several factors with Cox models.

Chronic palsy: The combination of surgical treatment is often necessary and standard for patients with chronic flaccid palsy. For instance, reconstruction of the facial nerve is often combined with eyelid suture. Here, indeed, we only used statistics, to evaluate the outcome. The subgroups become too small (Subgroups with N<10) to perform proper multivariate statistics. This is a clear limitation. In general, it is difficult to reach large sample sizes for specific subgroups for such kind of very individualized surgery. Therefore, we added some sentences in the Discussion to explain this limitation (lines 395 ff): “As it is typical for treatment of chronic facial palsy, most patients received combinations of different treatments (several types of surgery, combination of surgical with non-surgical treatment). Therefore, it was not feasible to perform next to the univariable analysis (focus on treatment) also to perform a multivariable analysis. The subgroups of identical treatment combinations were too small to allow a multivariable analysis. Therefore, we could not analyzed the interaction between different treatment types on the outcome in patients with chronic facial palsy.”

2) Figure 2 is too difficult to follow. Please find another way to present this information, or select the most important points of the figure and leave the current figure as a supplement. 

Answer #2.2: Yes, the details are very small and the reader had to zoom into the figure to see all details. The primary aim was to show at one glance that the treatment effects are different for different treatment types but also dependent one the outcome measure used. We reduced the number of graphs to six and shifted the complete figure to the Supplements. It is now Supplementary Figure S4.

3) The last part of section 3.5 is just an afterthought at this point and should be removed from the paper or significantly modified (lines 306-319)

Answer #2.3: Concerning 2.1, we do not understand why the reviewers labels the last part of 3.5 an “afterthought”. The presented study is a retrospective study with the typical limitations addressed in the Discussion (lines 386 ff). Actually, any retrospective analysis is completely an “afterthought”. We present results for both, patients with acute and chronic facial palsy. In 2.1, the reviewer #1 plausibly demanded to analyze the interaction between different types of therapy in the same patients on the outcome. This is what we did exactly in this paragraph for the patients with acute facial palsy. Therefore, we recommend leaving this part of section 3.5.

4) There are a lof of very long sentences that make the paper difficult to follow. Please proofread your document to clarify the text. 

Answer #1.4: The text was proofread again by a native speaker.

Some examples are:

  • (lines 60-64) As the patients are not only monitored by facial nerve grading but 62 from the beginning also by facial neve specific patient-related outcome measures 63 (PROMs), we will also present standardized outcome results.

Answer #1.4a: We revised this part, splitting the information into several shorter sentences: “The patients are not only monitored by facial nerve grading but from the beginning also by facial nerve specific patient-related outcome measures (PROMs). This allows us to present standardized outcome results.”

  • (lines 42-45) Although several clinical guidelines based on high evidence clinical trials have been published, still many acute cases do not receive any or optimal treatment.  Moreover, patients with low probability to recover or already occurred incomplete recovery are not referred or referred very late to specialized facial palsy services [2,6].

Answer #1.4b: Also here we shortened the sentences: “Although clinical guidelines have been published, still many cases do not receive optimal treatment. Moreover, patients with low probability to recover are not referred or referred very late to specialized facial palsy services”.

Orlando Guntinas-Lichius

for all authors

Jena, 17-Dec-2021

Reviewer 2 Report

Really interesting article and very well written. A research carried out in detail that will certainly be a point of reference for numerous colleagues and future systematic reviews.

1) Have you considered now or for the future a possible investigation related to the use of laser photobiomodulation to avoid the massive use of drugs for too long periods? ex: https://www.mdpi.com/1235942

2) Have you investigated dental iatrogenic damage? ex: https://www.mdpi.com/1081600

Author Response

Point-by-point response

Manuscript ID: jcm-1509047

Thank you very much for the detailed comments of the three reviewers. We answer all queries point-by-point.

Reviewer #2

Really interesting article and very well written. A research carried out in detail that will certainly be a point of reference for numerous colleagues and future systematic reviews.

1) Have you considered now or for the future a possible investigation related to the use of laser photobiomodulation to avoid the massive use of drugs for too long periods? ex: https://www.mdpi.com/1235942

Answer #2.1: Not yet. This was a retrospective analysis, hence it is clear that it is not possible to include post-hoc any new and innovative technology. We think that the data is too preliminary to become a standard alternative for clinical routine. Most clinicians do not yet know this innovative approach. As it is said in the article: “However, randomized controlled trials are necessary to sustain our encouraging results, exclude bias, and better explain the boundary between the time from diagnosis and the recovery of BP through PBM therapy.” – Definitively, we will follow the progress of photobiomodulation.

2) Have you investigated dental iatrogenic damage? ex: https://www.mdpi.com/1081600

Answer #2.2: Thanks to reference us to this review on “Mitochondrial Bioenergetic, Photobiomodulation and Trigeminal Branches Nerve Damage”. We checked the database again. None of the patients at a dental iatrogenic damage as cause.

Orlando Guntinas-Lichius

for all authors

Jena, 17-Dec-2021

Reviewer 3 Report

In general, the article is interesting, however, it requires introducing the indicated changes.

Many of the sentences are too general. These sentences need to be elaborated further. Examples:

„Focusing on the multidisciplinary team approach, the advantages for treatment of patients with Bell’s palsy [8], and the experience of a facial nerve unit with patients with facial palsy after skull base surgery were recently reviewed [5].”

“Other, more historical but large (>1,000 patients) series focused on Bell’s palsy [11-13]”

“The most recent report comes from the Sydney Facial Nerve Clinic on 145 patients treated between 2015 and 2018 [10]”

The authors do not describe the novelty of the study. They do not relate the purpose of the article to the existing literature.

The next too general sentences:

“Details on the diagnostics were published recently  [14,15]”

“Electrodiagnostic tests were a key component to evaluate the facial motor function and are described in detail elsewhere [16]”

The results of the statistical tests are not recorded according to scientific standards, e.g. Wilcoxon test: Z = 3.4; p = 0.02.

Each result of the statistical test applied should be recorded according to a scientific standard. Otherwise, the results are full of doubts. This is the main drawback of the article.

In the tables, the authors did not include information about the statistical test used.

Tables should be structured differently, so that it is known what comparisons the authors have made.

The authors should add the median statistic in the article.

Author Response

Point-by-point response

Manuscript ID: jcm-1509047

Thank you very much for the detailed comments of the three reviewers. We answer all queries point-by-point.

Reviewer #3

In general, the article is interesting, however, it requires introducing the indicated changes.

Many of the sentences are too general. These sentences need to be elaborated further. Examples:

„Focusing on the multidisciplinary team approach, the advantages for treatment of patients with Bell’s palsy [8], and the experience of a facial nerve unit with patients with facial palsy after skull base surgery were recently reviewed [5].”

Answer #3.1: We elaborated the background as recommended: The passage is now phrased as follows: “A multidisciplinary collaboration including a wide variety of subspecialties has proven effective for treatment of patients with Bell’s palsy [8]. A patient centred approach utilising physiotherapy, targeted botulinum toxin injection and selective surgical intervention offered by a multidisciplinary team can effectively reduce the burden of long-term disability for patients with Bell’s palsy and longstanding sequaelae [8]. The same has been shown for the management of facial paralysis following skull base surgery [5]. These patient profit from a multidisciplinary intervention, because an individualized combination of pharmacologic therapy, physical therapy for facial neuromuscular retraining, and surgical intervention is needed for most patients [5].”

“Other, more historical but large (>1,000 patients) series focused on Bell’s palsy [11-13]”

Answer #3.2: We elaborated the limitations of these studies now as recommended: “These historical series do not include standardized outcome measures and are limited to measurement of the physical dysfunction of the patients. These studies do not include data on quality of life of the patents or psychosocial dysfunction”

“The most recent report comes from the Sydney Facial Nerve Clinic on 145 patients treated between 2015 and 2018 [10]”

Answer #3.3: We elaborated the background as recommended: “The Sydney team also used both, classical facial grading and PROMs for initial assessment of the patients and during follow-up. This allowed to show that is not the physical level of function that a patient has, but the social and psychological impact of their palsy that drives them to presentation in a specialized facial nerve center”

The authors do not describe the novelty of the study. They do not relate the purpose of the article to the existing literature.

Answer #3.4: The more detailed explanations in the Introduction (see answer 3.1 to 3.3) allowed us a better transition to the aim of our study. We revised the last paragraph: “When assessing the outcome of a facial nerve center, it seems to be advisable to measure the outcomes with standardized grading tools and PROMs. This is of interest for both, patients with acute and chronic facial palsy. Therefore, we aim to report the experience from a German multidisciplinary facial nerve center, treating patients with acute and chronic facial palsy. The focus lies on the diagnostic and therapeutic management of 1,220 patients. The patients are not only initially assessed and later on monitored by facial nerve grading but from the beginning also by facial nerve specific (PROMs). This allows us to critically analyze the outcome in our center. Furthermore, we can compare the results to other multidisciplinary centers using a similar approach.”

The next too general sentences:

“Details on the diagnostics were published recently  [14,15]”

Answer #3.5: In the sentences following this sentence, we explain what kind of examinations of were performed in all patients. I think that it should not be the task of this article to explain standard diagnostic tests. We propose to delete this sentence.

“Electrodiagnostic tests were a key component to evaluate the facial motor function and are described in detail elsewhere [16]”

Answer #3.6: Actually, also here we explain already what kind of electrophysiological information was most relevant for this study. Reference 16 explains the methods in more detail. In accordance to the previous query (#3.5), we propose to delete this sentence

The results of the statistical tests are not recorded according to scientific standards, e.g. Wilcoxon test: Z = 3.4; p = 0.02. Each result of the statistical test applied should be recorded according to a scientific standard.  Otherwise, the results are full of doubts. This is the main drawback of the article. In the tables, the authors did not include information about the statistical test used. Tables should be structured differently, so that it is known what comparisons the authors have made. The authors should add the median statistic in the article.

Answer #3.7: Thank you very much for this attentive hint. The information given in the Methods was not correct. Please excuse this inattentiveness. We performed both, the Wilcoxon test and a paired t-test when comparing dependent data. The results concerning significant and non-significant differences between subgroups were identical for both methods. Finally, we decided to show the results from the paired t-test, because it is usual to present here the test results with the mean ± standard deviation values. The use of the different statistical tests is described in the Methods, section 2.5. Following the advice of reviewer #3, we added information of the used statistical test in all tables. Now, for each table the statistics behind should be clear. The same was done for the Supplement Tables. Here, we also added the information on the exact test used under each Supplement Table.

Orlando Guntinas-Lichius

for all authors

Jena, 17-Dec-2021

Round 2

Reviewer 3 Report

The article has been corrected, but the results are still not recorded according to scientific standards, e.g.:

a) Chi-square

X2 (degress of freedom, N = sample size) = chi-square statistic value, p = p value

b) ANOVA

F(1, 145) = 5.43, p < 0.001

c) paired t-test

t(20) = 8.36; p < 0.001

This would increase the value of the publication, i.e. its quality.  This is true for any statistical test used.

There is no information about post-hoc testing.

Round 3

Reviewer 3 Report

The authors have made sufficient changes, I accept the manuscript.